# Enantioselective Cyclopropanation Catalyzed by Gold(I)-Carbene Complexes

**DOI:** 10.3390/molecules27185805

**Published:** 2022-09-07

**Authors:** Zita Szabo, Sophia Ben Ahmed, Zoltan Nagy, Attila Paczal, Andras Kotschy

**Affiliations:** Servier Research Institute of Medicinal Chemistry, Záhony u. 7., 1031 Budapest, Hungary

**Keywords:** asymmetric synthesis, gold catalysis, carbene complexes, cyclopropanation

## Abstract

The formation of polysubstituted cyclopropane derivatives in the gold(I)-catalyzed reaction of olefins and propargylic esters is a potentially useful transformation to generate diversity, therefore any method in which its stereoselectivity could be controlled is of significant interest. We prepared and tested a series of chiral gold(I)-carbene complexes as a catalyst in this transformation. With a systematic optimization of the reaction conditions, we were able to achieve high enantioselectivity in the test reaction while the *cis:trans* selectivity of the transformation was independent of the catalyst. Using the optimized conditions, we reacted a series of various olefins and acetylene derivatives to find that, although the reactions proceeded smoothly and the products were usually isolated in good yield and with good to exclusive *cis* selectivity, the observed enantioselectivity varied greatly and was sometimes moderate at best. We were unable to establish any structure-property relationship, which suggests that for any given reagent combination, one has to identify individually the best catalyst.

## 1. Introduction

*N*-heterocyclic carbenes (NHC), since the first isolation of a metal-free imidazol-2-ylidene by Arduengo and co-workers in 1991 [1], have become one of the most extensively investigated ligands for transition metal complexes [2,3,4,5,6]. Compared to the corresponding phosphine ligands, NHC complexes have stronger metal−carbon σ-bonds and exhibit higher activities [7,8,9,10,11,12]. In addition, a diverse array of NHC with various ring sizes, ring backbones, heteroatoms, and N-substituents has been readily prepared [13]. Among those, the most versatile representative of exceedingly steady NHC is the five-membered NHC [14], owing to proximal σ-electron-withdrawing and π-electron-donating nitrogen atoms next to the carbene carbon atom.

However, while gold(I)-catalyzed intramolecular transformations usually perform outstandingly, their intermolecular versions are more challenging [15].

Gold-catalyzed reactions have emerged as a powerful synthetic tool in modern organic synthesis. This past decade saw a rapid increase in the number of published homogeneous gold-catalyzed transformations, fuelled by the advantages offered by gold complexes as catalysts [16]. In comparison to other transition-metal catalysts, most gold-catalyzed reactions are atom-economic, remarkably mild with regard to reaction conditions, and most importantly, have a different reaction scope [17,18]. Gold(I)-catalyzed reactions also offer a powerful tool for the construction of molecular complexity. Reactions of 1,6-enynes with electrophilic metal complexes as catalysts have been extensively studied in the last few decades, which has allowed for the determination of the main pathways that follow the coordination of the metal to the triple bond of these substrates [19,20,21].

Small cycloalkanes are favored building blocks in medicinal chemistry due to their well-defined conformation, which is achieved with an all-sp^3^ core that improves drug-like properties [22]. In spite of this interest, the stereocontrolled generation of polysubstituted cyclopropanes and cyclobutanes remains challenging. The transformation of easily accessible propargyl carboxylates and alkenes in the presence of gold(I) catalysts reported by Toste [23] attracted our attention because it yields trisubstituted cyclopropanes with a well-defined substitution pattern and a collection of substituents that are suitable for further orthogonal chemical transformations. Using chiral phosphine-gold complexes, they were able to achieve good yields (typically in the 60–80% range) and enantioselectivities (varying between 60% and 94%). Attempts to employ the more easily accessible chiral gold(I)-carbene complexes in this transformation remained only partially successful so far. Espinet and collaborators achieved only a little stereoselection (*ee* up to 24%) with acyclic carbene complexes [24]. By using an NHC-gold(I) complex, Strand and co-workers [25], (Figure 1) achieved similar enantioselectivity. Having recently reported the synthesis of a diverse set of chiral NHC precursors and their conversion to silver(I) and gold(I) complexes [26,27,28], we decided to investigate how the structure of the chiral NHC-gold(I) complex influences its performance in the asymmetric cyclopropanation reaction.

## 2. Results and Discussion

In the first stage of the study, we tested 42 gold(I)-*NHC*-complexes (**C1–C42**) in the reaction of the propargyl pivalate **1a** and propenyl acetate **2a**, which were prepared from the appropriate dihydroimidazolium salts through the respective silver(I)-carbene complex (**C1–4**, **C6–40**) or by direct synthesis (**C5**, **C41–42**) following standard protocols (Figure 2). For practical reasons, we slightly modified Strand’s conditions, decreasing the catalyst and co-catalyst loading to 2.5% and increasing the reaction temperature to −25 °C. The active catalysts were generated in situ by mixing the gold complexes with the silver salt at room temperature for 10 min, and the progress and selectivity of the reactions were monitored for up to 14 days by chiral HPLC. The reactions produced both *cis* and *trans* isomers, each as a mixture of enantiomers. The ratio of the formed *cis* and *trans* isomers was around 71:29 in every case and we were unable to improve this by optimizing the reaction parameters. The enantiomeric ratios of the *cis* and *trans* fractions were determined by chiral HPLC on a Lux 5 µm i-Amylose column. We did not determine the absolute stereochemistry of the products; enantiomers were only identified by their elution order. The results of the complex screen are collected in Table 1.

As a general conclusion, we established that the transformation was slow irrespective of the complex used. The earliest time point to reach full conversion was after 6 days for some complexes (e.g., entries 1–3, 5–6, and 36), while for some others (e.g., entries 10–11, 18, and 29), conversion was incomplete even after 2 weeks. The observed enantioselectivities varied in a broad range from no or very low selectivity (e.g., entries 2, 4, 6, 8–13, 41) to acceptable (e.g., entries 15, 19, 26, 30, 35–36, 38). The optimal combination of activity and selectivity was delivered by complex **C36,** reaching full conversion in 6 days and delivering enantioselectivities of 82% for both the *cis* and *trans* isomers.

It is interesting to observe the influence of certain structural features of the complex on the enantioselectivity of the reaction. For complex pairs **C1**–**C8** having the same *S*-backbone chirality and variation of a small (methyl) and larger (phenyl, 1-naphthyl, 2-naphthyl) substituent in the chiral center close to the gold coordination site, we observed that the *S*-enantiomers **C1**, **C3**, **C5**, and **C7** are more efficient (*c.f.* 36–58% *ee* for **3a** in entries 1, 3, 5, and 7, compared to 10–14% *ee* in entries 2, 4, 6, and 8). The major *cis* enantiomer is the same for the efficient complexes. In a consistent manner, switching the backbone chirality to *R* and having an *S*- or *R*-adamantylethyl substituent on the NHC (entries 13–14). The more efficient complex was **C14,** preferring the formerly minor *cis* enantiomer (*c.f*. entries 7 and 14).

When the only chirality element of the complex is on the distant backbone (**C9**, **C10**), not surprisingly, the induction is negligible. It is also worth mentioning that we observed a significant drop in the activity of the complexes (entries 9, 10). From our structural studies of these gold-NHC complexes, we knew [26] that the backbone chirality element exerts a significant influence on the orientation of the adjacent ortho-substituted benzene ring; therefore, we replaced the *o,o’*-disubstituted ring with the *o*-*tert*-butyl benzene (**C11–12**). Maintaining the *S*-phenethyl substitution and varying the backbone chirality had little effect on the outcome. Both catalysts showed a decreased reactivity and enantioselectivity (entries 11, 12).

The next set of gold complexes had a sterically more demanding, homochiral 1,2-diarylethyl substituent attached to the ring nitrogen (**C15**–**C38**). The epimeric compound pairs **C15**–**C16**, **C17**–**C18**, and **C19**–**C20** were explored to determine if the electron density of the proximal benzene ring has an effect on the efficiency of the complex. Irrespective of the variations of the substituent, we observed the same trend and very similar selectivity, which depended only on the substituent’s chirality compounds in the R-series, consistently leading to a superior result (*c.f.* entries 15, 17, 19, and 16, 18, 20). Moving the benzene substituent from the para to the ortho position (**C21**–**C24**) had a systematic effect. The selectivity in the previously more efficient *R*-series dropped considerably (*cf.* 30% for **C21** vs. 70% for **C17**, 38% for **C23** vs. 70% for **C19**) while in the S-series we observed an opposite trend (*cf.* 52% for **C22** vs. 40% for **C18**, 52% for **C24** vs. 44% for **C20**). The enantiomer preference remained the same in both series. These results also underline that steric factors in this region play an important role in determining the catalyst’s activity. This is also supported by the observation that complexes bearing the methoxy substituent in the 3-position (**C25**, **C26**) showed a very similar efficiency to the 4-methoxy analogues (*cf.* entries 17 and 25, as well as entries 18 and 26).

Substitution of the distal benzene ring of the 1,2-diarylethyl substituent (**C27**–**C32**) showed a similar structure-efficiency relationship to the proximal one. Complexes with a 4-substituted ring behaved alike irrespective of the electronic nature of the substituent (*cf.* entries 27 and 29, as well as entries 28 and 30) while moving the substituent t into the 2-position led to a deterioration for the formerly more efficient complex (58% for **C31** vs. 76% for **C29**) and improvement for the formerly less efficient complex (62% for **C32** vs. 34% for **C30**). Moving the chlorine substituent into the 3-position gave the most efficient complexes. **C35** and **C36** led to a full conversion in 7 and 6 days, respectively, and achieved enantioselectivities of 74% and 82% for the major *cis* isomer, as well as 66% and 82% for the minor *trans* isomer, respectively. In *N.B.,* the reversal of the preferred enantiomer is due to the inversion of the backbone chirality to *R* from the *S*, present in the complexes discussed so far.

We also prepared the complexes that had a 3-trifluoromethyl substituent in the proximal benzene ring (**C37**, **C38**). This change, in line with previous observations, gave acceptable but slightly inferior results (*c.f.* 54% vs. 74% and 70% vs. 82%, respectively). To assess if the backbone chirality plays any role in this catalytic process, we also prepared the des-tert-butyl analogues of **C29** (**C39**) and **C30** (**C40**). While in the **C29**-**C39** pair, the former showed superior selectivity (76% vs. 48% and 72% vs. 46%), we also observed a reversal of the preferred enantiomer. Naturally, the reversal of the enantiomer preference was also observed for the **C30**–**C40** complex pair, but here the latter complex was the more efficient one (34% vs. 60% and 24% vs. 54%). We have also tested the gold complexes **C41** and **C42**, the former being the most efficient complex reported by Strand and the latter its backbone substituted analogue. In line with the published results, the selectivity obtained with **C41** was around 20%. The introduction of the backbone chirality element (entry 42) was advantageous, similarly to the previous observations, leading to enantiomeric excesses of 36% and 38%, respectively.

After selecting the gold complexes **C35** and **C36** for further screening, we studied the effect of the silver salt (Table 2). In the absence of a silver salt, we observed no transformation with either complex (entry 3). Replacing AgNTf_2_ with AgSbF_6_, on the other hand, led to complete conversion after 1 day and conserved the selectivity (entries 4, 5). Changing the anion to tetrafluoroborate (entries 6, 7) gave similar results to the original system, while using the hexafluorophosphate salt (entries 8, 9) deteriorated both the activity and selectivity of the catalytic system. In the presence of coordinating anions tosylate (entries 10, 11), acetate (entry 12), trifluoroacetate (entry 13), and camphorsulfonate (entry 14) we also observed diminished reactivity and selectivity.

In parallel, we also studied the effect of the solvent on the transformation (Table 3). Replacing DCM with other chlorinated solvents (entries 2, 3) had little effect on the selectivity of the transformation, but the reaction was significantly slower in chloroform than in dichloroethane. The use of aprotic solvents of varying polarity (entries 4–8) led to decreased catalyst activity and selectivity, while we observed no conversion using methanol or acetone as solvent (entries 9, 10). Interestingly, changing the solvent to trifluoroethanol (entries 11, 12) led to a significant acceleration of the transformation and a moderate improvement of the selectivity with both catalysts. Switching to hexafluoroisopropanol (entries 13, 14), we had to increase the reaction temperature to 0 °C due to its high freezing point, and the obtained results were very similar to trifluoroethanol. With both solvents, we achieved full conversion within one day and the enantioselectivity of the formed products was in the 74–86% range.

Having identified TFE and HFIPA as efficient solvents and AgSbF_6_ as an optimal silver salt, we also studied their combined effect at various temperatures and catalyst loadings (Table 4). First, we ran the reaction in the presence of 2.5 mol% catalyst **C35** and AgSbF_6_ between −25 °C and 25 °C (entries 1–3) and saw that the time required to reach full conversion decreased from 2.5 h to 15 min. Unfortunately, it was also accompanied by a concomitant decrease in selectivity. We observed very similar changes with 2.5% catalyst **C36** in the −40 °C–25 °C temperature range (entries 4–7). Running the reaction at −40 °C increased the time needed to reach completion to 17 h but also gave the highest enantioselectivities in this series. We also repeated the experiments in HFIPA at 0 °C and 25 °C with both catalysts at 2.5% loading (entries 8–9 and 10–11). The increase in the reaction temperature accelerated the transformation, but we also observed a small but perceptible erosion of the selectivity in both cases.

In the next set of experiments, we systematically varied the catalyst loading and studied its effect at different temperatures in the two solvents. Using catalyst **C35** in TFE at −25 °C the decrease of its loading to 1% and 0.5% led to increased reaction times and a slight decrease in selectivity (*c.f.* entries 1, 12, and 13). A similar effect was observed for **C36** in TFE both at −40 °C (*c.f.* entries 4, 14) and at −25 °C (*c.f.* entries 5, 15, and 16). Switching the solvent to hexafluoroisopropanol we ran reactions with decreasing loading of **C35** (entries 17–19) and **C36** (entries 20–22) at 0 °C. While the reaction rate decreased with the catalyst loading, the stereoselectivity of the transformation remained similar for both catalysts. It is important to note that the slight decrease in selectivity observed for entry 13 might arise from the less specific but faster transformation of the unreacted materials during workup. When we ran the reaction with 1% of **C36** in the 1:1 mixture of the two solvents at −25 °C (entry 23), we found that the reaction time was similar to that in TFA alone (24 h as in entry 12), but the enantioselectivity of the cis and trans products was 92% and 88%, respectively, the highest values observed so far.

Upon completion of the optimization studies, we went back to our starting set of chiral carbene complexes and with a selection including **C3**, **C4**, **C15**, **C16**, **C21**, **C22**, **C25**, **C26**, **C31**, and **C32** re-run the test reaction. We observed a significant increase in catalytic activity in all cases and an improved selectivity for most catalysts (for details see the Appendix A).

On completion of the optimization, we tested the scope of the transformation by reacting a selection of substituted propargyl pivalates (**1a–d**) with different olefins (**2a–q**) in the presence of 1 mol% **C36** as a catalyst and 1 mol% silver hexafluoroantimonate as an additive in the 1:1 mixture of trifluoroethanol and hexafluoroisopropanol at −25 °C. Although in this system the optimization studies showed a complete conversion in 2 days, we ran our experiments for 5 days to enable full conversion in the case of an unexpectedly slow reacting substrate (N.B. on warming up during workup, one can incur fast and non-selective transformation of the unreacted starting material). On workup of the reaction mixture, we determined the ratio of the cis (**3**) and trans (**4**) products, as well as their enantioselectivity by chiral HPLC, then isolated the products using flash chromatography, and finally re-checked the enantiomeric excess of the products. The results are presented in Table 5. In general, we can conclude that the transformation proceeded readily in most cases, leading to the isolation of the products in good to moderate yield. We can also conclude that, typically, we observed a preference for the formation of the cis product (**3a–p**).

In the first set of experiments, we varied the substitution of the propargylic reagent (**1a–c**, entries 1–3) and observed a very similar **3**:**4** selectivity (around 7:3) in each case. For 1a and 1b, the enantioselectivities were similarly high, while for 1c we saw a decrease in selectivity for both **3c** and **4c**. The next set of reactions included **1a** and the olefins **2b,c**, which contain a sterically less hindered double bond (entries 4–5). On one hand, we saw a shift of the **3**:**4** ratio to around 55:45, accompanied by a slight decrease in the enantioselectivity of both products.

In the following transformations, **1a** was reacted with a collection of olefins (**2d–i**), in which an aromatic substituent was directly attached to the double bond (entries 6–11). All reactions proceeded with high, sometimes exclusive, cis selectivity, and the products were usually isolated in good yield. Unfortunately, we have also observed a steady erosion of the stereoselectivity of these transformations. It was difficult to draw any more specific structure-property relationship on the basis of the obtained data. Neither the position nor the electron withdrawing-donating nature of the substitution correlated systematically with the observed *ee* values.

Switching to E-1,2-disubstituted olefins (**2j–l**, entries 12–14) we made similar observations. β-methylstyrene (**2j**) gave a mixture of the cis and trans products, both with low to moderate enantioselectivity. Increasing the steric bulk of the substituent to stilbene (entry 13), we only observed the formation of the cis product and stereoselectivity was mediocre again. Similar results were obtained when we replaced one of the phenyl rings with a more elaborate aliphatic substituent (entry 14). In the next experiment, we used the 1,1-disubstituted olefin **2m** (entry 15), which is a close analogue of **2a**. We observed a 67:33 *cis*:*trans* ratio with good (75% and 65% respectively) yields and reasonable (68% and 70%) stereoselectivity (*c.f.* 71:29 ratio, 68% and 42% yields, and 92% and 88% *ee*s for **2a**), underlining the connection between the olefin’s substitution pattern and the selectivity of the transformation. When we used the cis-1,2-disubstituted olefin, cyclohexene (**2n**, entry 16), we observed exclusive *cis*-selectivity and moderate enantioselectivity while the product was isolated in mediocre yield.

In the last set of experiments (entries 17–20), we varied the substitution of the acetylene reagent. The reaction of the ethynyl-cyclohexane **1b** with acetylstyrene (**2o**) followed the previously observed trends (*c.f.* entries 7, 10). The reaction proceeded with an 82:18 cis-selectivity and moderate enantioselectivity. In this case, we were unable to separate the cis and trans isomers by column chromatography, so the product was isolated and characterized as a mixture. The next acetylene reagent tested was the phenylpropargyl pivalate **1c**, which was reacted with vinylcyclopentane (**2p**). Although the reaction proceeded smoothly and both the *cis* and *trans* products were isolated in acceptable yield, the enantioselectivity of the transformation was very poor. Finally, we tested the phenylpropargylic acetate **1d** in combination with styrene derivatives **2f** and **2q** (entries 19, 20). With bromostyrene **2f,** we observed similar results to its reaction with **1a** (*c.f.* entry 8). The yield of the transformation was good, but there was no sign of enantiomer differentiation. The combination of the iodostyrene **2q** and **1d** (entry 20) led to poor yield and low enantioselectivity, probably the least advantageous result of the whole study. For these last two transformations, the products were isolated as a mixture of the cis and trans isomers.

## 3. Discussion

We prepared over 40 chiral gold(I)-carbene complexes and tested them in the reaction of propargylic esters and various olefins, yielding polysubstituted cyclopropane derivatives. We established that the *cis:trans* selectivity of the transformation was determined primarily by the substitution pattern of the reagents, while the catalyst’s structure influenced the enantioselectivity of the transformation. With the most promising catalysts, we studied the influence of the applied conditions on the transformation. Of the silver salts used as an additive, silver hexafluoroantimonate gave the best results. In a solvent screen, the fluorinated alcohols, trifluoroethanol and hexafluoroisopropanol stood out, shortening the reaction time considerably and maintaining good selectivity at elevated temperatures. With these optimized conditions, we were able to decrease the catalyst and additive loading without compromising the efficiency of the transformation. Finally, using the optimized conditions, we tested the scope of the transformation. We found that the gold(I)-catalyzed cyclopropanation proceeded smoothly and with a good yield for most reagent combinations. The formation of the *cis* product was preferred in all cases, sometimes to the extent of being the exclusive product in the reaction. Unfortunately, the observed enantioselectivity varied in a wide range depending on the reagents, suggesting that for a given pair of reagents, one will need to re-run the catalyst selection and optimization to achieve good stereoselectivity. While this finding is disappointing, the structural complexity of the obtained products might justify the repeated catalyst selection process for specific target molecules of interest.

## 4. Materials and Methods

General information on the reported syntheses as well as the detailed characterization of all prepared compounds is provided in the Appendix A.

## Figures and Tables

**Figure 1 molecules-27-05805-f001:**
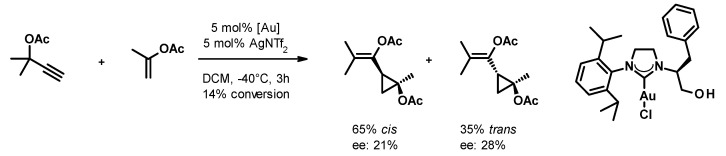
Asymmetric cyclopropanation reaction.

**Figure 2 molecules-27-05805-f002:**
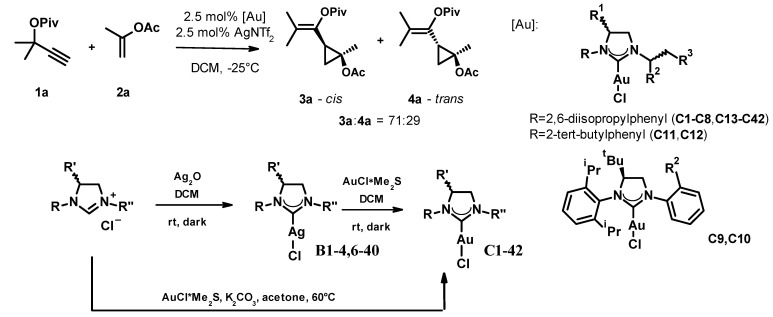
The optimization of the asymmetric cyclopropanation reaction and the general synthesis scheme of the used catalysts.

**Table 1 molecules-27-05805-t001:** Catalyst screening (**C1–42**) in the asymmetric cyclopropanation.

Entry	[Au]	R^1^	R^2^	R^3^	3a *ee* (%)	4a *ee* (%)	Time (d) ^1^
**1**	**C1**	*S*-^t^Bu	*S*-Ph	H	50 ^2^	44	6
**2**	**C2**	*S*-^t^Bu	*R*-Ph	H	14 ^2^	12	6
**3**	**C3**	*S*-^t^Bu	*S*-1-naphthyl	H	36 ^2^	32	6
**4**	**C4**	*S*-^t^Bu	*R*-1-naphthyl	H	14	14 ^2^	8
**5**	**C5**	*S*-^t^Bu	*S*-2-naphthyl	H	46 ^2^	44	6
**6**	**C6**	*S*-^t^Bu	*R*-2-naphthyl	H	12 ^2^	12	6
**7**	**C7**	*S*-^t^Bu	*S*-^t^Bu	H	58 ^2^	48	7
**8**	**C8**	*S*-^t^Bu	*R*-^t^Bu	H	10	0	9
**9**	**C9**	*S*-^t^Bu	2-isopropylphenyl		12 ^2^	16	8
**10**	**C10**	*S*-^t^Bu	2-tertbutylphenyl		2 ^2^	20	14 (85)
**11**	**C11**	*S*-^t^Bu	*S*-Ph	H	12 ^2^	10	14 (84)
**12**	**C12**	*R*-^t^Bu	*S*-Ph	H	6 ^2^	2	14 (98)
**13**	**C13**	*R*-^t^Bu	*S*-adamantyl	H	26 ^2^	4	9
**14**	**C14**	*R*-^t^Bu	*R*-adamantyl	H	50	42 ^2^	10
**15**	**C15**	*S*-^t^Bu	*R*-4-CF_3_-Ph	Ph	74 ^2^	64	10
**16**	**C16**	*S*-^t^Bu	*S*-4-CF_3_-Ph	Ph	44	42 ^2^	9
**17**	**C17**	*S*-^t^Bu	*R*-4-MeO-Ph	Ph	70 ^2^	64	7
**18**	**C18**	*S*-^t^Bu	*S*-4-MeO-Ph	Ph	40	42 ^2^	14 (82)
**19**	**C19**	*S*-^t^Bu	*R*-4-Me-Ph	Ph	70 ^2^	64	7
**20**	**C20**	*S*-^t^Bu	*S*-4-Me-Ph	Ph	44	42 ^2^	8
**21**	**C21**	*S*-^t^Bu	*R*-2-MeO-Ph	Ph	30 ^2^	28	10
**22**	**C22**	*S*-^t^Bu	*S*-2-MeO-Ph	Ph	52	48 ^2^	10
**23**	**C23** ^3^	*S*-^t^Bu	*R*-2-Me-Ph	Ph	38 ^2^	32	14
**24**	**C24**	*S*-^t^Bu	*S*-2-Me-Ph	Ph	52	54 ^2^	14
**25**	**C25**	*S*-^t^Bu	*R*-3-MeO-Ph	Ph	70 ^2^	64	8
**26**	**C26**	*S*-^t^Bu	*S*-3-MeO-Ph	Ph	36	42 ^2^	9
**27**	**C27**	*S*-^t^Bu	*R*-Ph	4-Me-Ph	68 ^2^	62	10
**28**	**C28**	*S*-^t^Bu	*S*-Ph	4-Me-Ph	38	46 ^2^	14
**29**	**C29**	*S*-^t^Bu	*R*-Ph	4-Cl-Ph	76 ^2^	72	8
**30**	**C30**	*S*-^t^Bu	*S*-Ph	4-Cl-Ph	34	24 ^2^	14 (91)
**31**	**C31**	*S*-^t^Bu	*R*-Ph	2-Cl-Ph	58 ^2^	56	7
**32**	**C32**	*S*-^t^Bu	*S*-Ph	2-Cl-Ph	62	58 ^2^	7
**33**	**C33**	*R*-^t^Bu	*R*-Ph	Ph	50 ^2^	46	9
**34**	**C34**	*R*-^t^Bu	*S*-Ph	Ph	64	68 ^2^	8
**35**	**C35**	*R*-^t^Bu	*R*-Ph	3-Cl-Ph	74 ^2^	66	7
**36**	**C36**	*R*-^t^Bu	*S*-Ph	3-Cl-Ph	82	82 ^2^	6
**37**	**C37**	*R*-^t^Bu	*R*-3-CF_3_-Ph	Ph	54 ^2^	46	8
**38**	**C38**	*R*-^t^Bu	*S*-3-CF_3_-Ph	Ph	70	74 ^2^	8
**39**	**C39**	H	*R*-Ph	4-Cl-Ph	48	46 ^2^	9
**40**	**C40**	H	*S*-Ph	4-Cl-Ph	60 ^2^	54	9
**41**	**C41**	H	*S*-hydroxymethyl	Ph	18	20 ^2^	10
**42**	**C42**	*S*-^t^Bu	*S*-hydroxymethyl	Ph	36	38 ^2^	10

^1^ Time needed to reach full conversion. Conversion values in parenthesis were reached after 14 days); ^2^ The enantiomer eluting second under the applied chromatographic conditions is the major one; ^3^ The cis:trans ratio was 6:4.

**Table 2 molecules-27-05805-t002:**
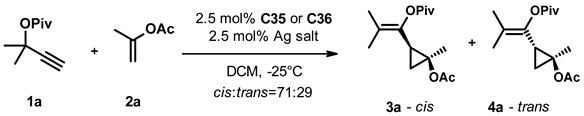
The effect of the added silver salt on the asymmetric cyclopropanation.

Entry	Complex	Ag Salt	3a *ee* (%)	4a *ee* (%)	Time (d) ^1^
**1**	**C35**	AgNTf_2_	74 ^2^	66	7
**2**	**C36**	AgNTf_2_	82	82 ^2^	6
**3**	**C35** or **C36**	none			no reaction
**4**	**C35**	AgSbF_6_	74 ^2^	68	1
**5**	**C36**	AgSbF_6_	80	78 ^2^	1
**6**	**C35**	AgBF_4_	70 ^2^	62	7
**7**	**C36**	AgBF_4_	78	78 ^2^	2
**8**	**C35**	AgPF_6_	nd		8 (33%)
**9**	**C36**	AgPF_6_	56	58 ^2^	8 (94%)
**10**	**C35**	AgOTs	nd		8 (32%)
**11**	**C36**	AgOTs	48	68 ^2^	8 (37%)
**12**	**C35** or **C36**	AgOAc			no reaction
**13**	**C35** or **C36**	AgTFA			no reaction
**14**	**C35** or **C36**	Ag-*S*-CSA or Ag-*R*-CSA	nd		8 (28–38%)

^1^ Time needed to reach full conversion. Conversion values in parenthesis were reached after 8 days; ^2^ The enantiomer eluting second under the applied chromatographic conditions is the major one.

**Table 3 molecules-27-05805-t003:**
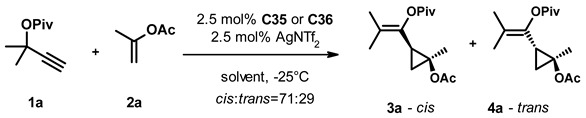
The effect of the solvent on the asymmetric cyclopropanation.

Entry	Complex	Solvent	3a *ee* (%)	4a *ee* (%)	Time (d) ^1^
**1**	**C35**	DCM	74 ^2^	66	7
**2**	**C36**	AgNTf_2_	82	82 ^2^	6
**3**	**C35**	chloroform	76 ^2^	62	5 (32%)
**4**	**C35**	DEE	62 ^2^	50	5 (30%)
**5**	**C35**	THF	56 ^2^	52	5 (30%)
**6**	**C35**	MeCN	40 ^2^	44	5 (21%)
**7**	**C35**	EtOAc	58 ^2^	48	5 (18%)
**8**	**C35**	toluene	64 ^2^	38	5 (20%)
**9**	**C35**	MeOH			no reaction
**10**	**C35**	acetone			no reaction
**11**	**C35**	TFE	82 ^2^	74	1
**12**	**C36**	TFE	84	78 ^2^	1
**13**	**C35**	HFIPA (0 °C)	84 ^2^	78	1
**14**	**C36**	HFIPA (0 °C)	86	82 ^2^	1

^1^ Time needed to reach full conversion. Conversion values in parenthesis were reached after 5 days; ^2^ The enantiomer eluting second under the applied chromatographic conditions is the major one.

**Table 4 molecules-27-05805-t004:**
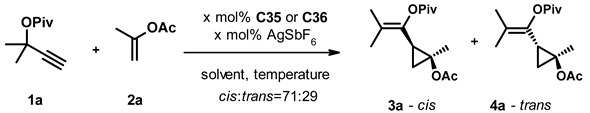
Fine-tuning the catalyst loading and the reaction temperature in the asymmetric cyclopropanation.

Entry	Complex (mol%)	Solvent	Temperature	3a *ee* (%)	4a *ee* (%)	Time (h) ^1^
**1**	**C35** (2.5%)	TFE	−25 °C	84 ^2^	82	2, 5
**2**	**C35** (2.5%)	TFE	0 °C	78 ^2^	70	0, 5
**3**	**C35** (2.5%)	TFE	25 °C	72 ^2^	70	0, 25
**4**	**C36** (2.5%)	TFE	−40 °C	88	84 ^2^	17
**5**	**C36** (2.5%)	TFE	−25 °C	84	78 ^2^	2, 5
**6**	**C36** (2.5%)	TFE	0 °C	84	78 ^2^	0, 5
**7**	**C36** (2.5%)	TFE	25 °C	78	70 ^2^	0, 25
**8**	**C35** (2.5%)	HFIPA	0 °C	84 ^2^	78	1, 5
**9**	**C35** (2.5%)	HFIPA	25 °C	78 ^2^	64	0, 5
**10**	**C36** (2.5%)	HFIPA	0 °C	84	80 ^2^	1
**11**	**C36** (2.5%)	HFIPA	25 °C	82	76 ^2^	0, 25
**12**	**C35** (1%)	TFE	−25 °C	84 ^2^	72	24
**13**	**C35** (0.5%)	TFE	−25 °C	80 ^2^	66	48 (34%)
**14**	**C36** (1%)	TFE	−40 °C	88	84 ^2^	96
**15**	**C36** (1%)	TFE	−25 °C	86	82 ^2^	21
**16**	**C36** (0.5%)	TFE	−25 °C	84	76 ^2^	48
**17**	**C35** (1%)	HFIPA	0 °C	84 ^2^	80	12
**18**	**C35** (0.5%)	HFIPA	0 °C	84 ^2^	80	21
**19**	**C35** (0.1%)	HFIPA	0 °C	86 ^2^	78	48
**20**	**C36** (1%)	HFIPA	0 °C	88	84 ^2^	4
**21**	**C36** (0.5%)	HFIPA	0 °C	86	84 ^2^	21
**22**	**C36** (0.1%)	HFIPA	0 °C	86	80 ^2^	24
**23**	**C36** (1%)	TFE:HFIPA (1:1)	−25 °C	92	88 ^2^	24

^1^ Time needed to reach full conversion; ^2^ The enantiomer eluting second under the applied chromatographic conditions is the major one.

**Table 5 molecules-27-05805-t005:**
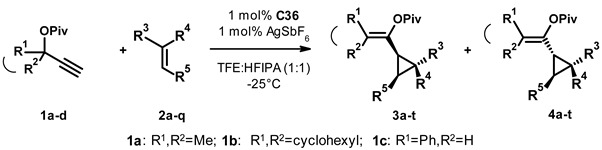
Scope and limitation of the asymmetric cyclopropanation using the optimized conditions.

Entry	Reagents		R^3^	R^4^	R^5^	3:4 ^1^	Products (Yield ^2^, *ee*)	
**1**	**1a**	**2a**	Me	OAc	H	71:29	**3a** (68%, 92 ^3^)	**4a** (42%, 88 ^3^)
**2**	**1b**	**2a**	Me	OAc	H	68:32	**3b** (68%, 90 ^3^)	**4b** (60%, 84 ^3^)
**3**	**1c**	**2a**	Me	OAc	H	73:27	**3c** (76%, 56)	**4c** (88%, 64)
**4**	**1a**	**2b**	C_2_H_4_CO_2_Me	H	H	54:46	**3d** (58%, 77)	**4d** (64%, 68)
**5**	**1a**	**2c**	C_2_H_4_-N-phthaloyl	H	H	57:43	**3e** (58%, 77)	**4e** (50%, 75 ^3^)
**6**	**1a**	**2d**	3-thienyl	H	H	100:0	**3f** (66%, 56)	
**7**	**1a**	**2e**	4-ClPh	H	H	80:20	**3g** (82%, 26)	**4g** (78%, 24)
**8**	**1a**	**2f**	3-BrPh	H	H	83:17	**3h** (72%, 0)	**4h** (66%, 0)
**9**	**1a**	**2g**	2-MePh	H	H	75:25	**3i** (76%, 46)	**4i** (62%, 16)
**10**	**1a**	**2h**	4-NO_2_Ph	H	H	87:13	**3j** (55%, 35)	**4j** (65%, 17)
**11**	**1a**	**2i**	Ph	H	H	100:0	**3k** (80%, 30)	
**12**	**1a**	**2j**	Ph	H	Me	74:26	**3l** (87%, 16)	**4l** (83%, 56)
**13**	**1a**	**2k**	Ph	H	Ph	100:0	**3m** (52%, 44)	
**14**	**1a**	**2l**	CH_2_CO_2_Me	H	Ph	100:0	**3n** (35%, 64)	
**15**	**1a**	**2m**	Ph	OAc	H	67:33	**3o** (75%, 68)	**4o** (65%, 70 ^3^)
**16**	**1a**	**2n**	H	-(CH_2_)_4_-		100	**3p** (52%, 44)	
**17**	**1b**	**2o**	4-AcPh	H	H	82:18	**3q (-,80) + 4q (-,46) 43% 4**	
**18**	**1c**	**2p**	cyclopentyl	H	H	67:33	**3r** (65%, 4 ^3^)	**4r** (53%, 7 ^3^)
**19**	**1d**	**2f**	3-BrPh	H	H	65:34	**3s** (-,0) + **4s** (-,0) 55% ^4^	
**20**	**1d**	**2q**	2-IPh	H	H	78:22	**3t** (-,6) + **4t** (-,28 ^3^) 25% ^4^	

^1^ The ratio of the *cis* and *trans* products in the crude reaction mixture; ^2^ Isolated yield based on the maximal achievable product derived from the *cis*:*trans* ratio of the crude product; ^3^ The enantiomer eluting second under the applied chromatographic conditions is the major one.; ^4^ The product was isolated as a mixture of cis and trans isomers.

## Data Availability

Not applicable.

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
