# Peer review of "Enantioselective Cyclopropanation Catalyzed by Gold(I)-Carbene Complexes"

_molecules, 2022, doi:10.3390/molecules27185805_

Round 1

Reviewer 1 Report

Szabo et al. report here a systematic study of gold-catalyzed cyclopropanation reactions between alkenes and propargylic esters.  Their main interest is in studying the applicability of chiral gold carbene complexes in this reaction, and in attempting to optimize the cis/trans selectivity and enantioselectivity of the reaction by varying the catalyst structure.  Unfortunately, while the yields are moderate-to-good, the authors had limited success in controlling either the cis/trans selectivity or the enantioselectivity, which was fairly poor in most cases and seem to mostly depend on the substrate structure rather than the catalyst.  Nevertheless, this is an extremely thorough study where the authors have characterized and tested an impressively large array of gold catalysts. The results will certainly be of interest to others working in the general areas of gold catalysis, cyclopropanations, carbene catalysis in general, etc.  I have no major issues with the paper but I do have a few very minor suggestions:

1. The fourth paragraph of the intro (Over the past two decades...) is quite redundant to the previous paragraph, I would omit it. 

2. In the scheme in Figure 2, it would be helpful to use some convention for the depictions of the silver and gold complexes that shows more clearly the carbene nature of the ligand/interactions.  Certainly it's obvious enough to experts but could be confusing to non-experts. 

3. Also in the scheme in Figure 2, I could not figure out what the * in "Me2S*AuCl" means.

4. On page 3 line 105 there appears to be a broken reference

5. On pg. 7 in the first paragraph there are a couple typos where it says AgSF6 instead of AgSbF6

6. On pg. 8 line 235, and again on pg. 10 line 309, I believe it is a mistake that the paper says "silver hexafluorophosphate" and it should actually say "silver hexafluoroantimonate" in these sentences.

7. I understand it may be impractical to put structures of all the ligands in the main text, but to be friendly to the reader I would suggest putting a chart of all the ligands and their corresponding numbers near the beginning of the SI, so that people don't have to keep scrolling all up and down the SI to see what ligands you're referring to at different points. 

Author Response

We thank the thorough refereeing of our manuscript. Our responses and actions are listed below.

1. The fourth paragraph of the intro (Over the past two decades...) is quite redundant to the previous paragraph, I would omit it. - this paragraph was merged with the following and repetitions were removed.

2. In the scheme in Figure 2, it would be helpful to use some convention for the depictions of the silver and gold complexes that shows more clearly the carbene nature of the ligand/interactions.  Certainly it's obvious enough to experts but could be confusing to non-experts. - we agree that the way carbene complexes are usually presented can be misleading for those less acquainted with the field so we chose another representation.

3. Also in the scheme in Figure 2, I could not figure out what the * in "Me2S*AuCl" means. - the formula was corrected to AuCl*Me2S

4. On page 3 line 105 there appears to be a broken reference - corrected

5. On pg. 7 in the first paragraph there are a couple typos where it says AgSF6 instead of AgSbF6 - corrected

6. On pg. 8 line 235, and again on pg. 10 line 309, I believe it is a mistake that the paper says "silver hexafluorophosphate" and it should actually say "silver hexafluoroantimonate" in these sentences. - corrected

7. I understand it may be impractical to put structures of all the ligands in the main text, but to be friendly to the reader I would suggest putting a chart of all the ligands and their corresponding numbers near the beginning of the SI, so that people don't have to keep scrolling all up and down the SI to see what ligands you're referring to at different points.  - a section was added to the SI

Reviewer 2 Report

Summary:

The report by Paczal, Kotschy et al. describes the optimization of enantioselective asymmetric cyclopropanation reactions using various olefins and propargylic esters, catalyzed via gold(I) carbene complexes. The authors first tested 42 synthesized gold(I) complexes as catalysts in the reaction of propargyl pivalate and propenyl acetate, showing these reactions occurred slowly and yielded poor stereoselectivity; catalyst structure was found to influence enantioselectivity. The most promising catalysts (C35 and C36) were then subjected to optimization. The most effective silver salt additive was found to be silver hexafluorophosphate, and the most effective solvents were trifluoroethanol and hexafluoroisopropanol, shortening reaction times considerably while maintaining good selectivity at elevated temperatures. These conditions allowed for a decrease in catalyst and additive loading without negatively impacting efficiency. Using these optimized conditions, the authors reacted various olefins and acetylene derivatives, finding that products were usually isolated in good yield and had good to exclusive cis selectivity. Enantioselectivity however, varied greatly and was often moderate (at best). No structure-property relationships were established, meaning for any given reagent combination one must individually identify the best catalyst. The work is interesting, though not ground-breaking. To this point, I could not help but feel underwhelmed after reading this paper. The authors should provide more reason as to why this is interesting and important… Notwithstanding, it does represent an advance and ought to be accepted for publication following a serious round of editing (grammar, language, sentence structure).

  • Page 1 line 7,8 in abstract: “olefins and propargylic esters is a potentially useful transformation to generate diversity therefore any method, in which its stereoselectivity could be controlled is of significant interest.” Should be changed to: “olefins and propargylic esters is a potentially useful transformation to generate diversity, therefore any method in which its stereoselectivity could be controlled is of significant interest.” Just move the comma in this sentence to fox the grammar. 
  • Page 1 line 10: “tested a series of chiral gold(I)-carbene complexes as catalyst I this transformation.” Should be corrected to “tested a series of chiral gold(I)-carbene complexes as catalysts in this transformation.”
  • Page 1 line 11: Change to “With a systematic optimization of the reaction conditions” 
  • Page 2 line 47: “ This generated a so far only partially met need for the stereo controlled generation of polysubstituted cyclopropanes and cyclobutanes” seems a bit wordy maybe change this up a bit. 
  • Page 2 line 57/58: “By using an NHC-gold(I) complex Strand and co-workers [18]. (Figure 1) achieved similar enantioselectivity.” Should be changed to “By using an NHC-gold(I) complex Strand and co-workers [18], (Figure 1) achieved similar enantioselectivity.
  • Page 2 line 73: “The progress of the reactions was monitored up to 14 days.” How were these reactions monitored? 
  • Page 2 line 78/79: “We did not determine the absolute stereochemistry of the isomers and enantiomers were only identified by their elution order.” Could be changed to “We did not determine the absolute stereochemistry of the isomers and enantiomers, they were only identified by their elution order.”
  • Page 3 line 100: End sentence after “(entries 13-14)” and start a new one it was confusing to read. 
  • Page 3 line 105: “ERROR! Bookmark not defined” was written fix this. 
  • Page 3 line 112-114: “Epimeric compound pairs C15-C16C17-C18C19-C20, explored if the electron density of the proximal benzene ring has an effect on the efficiency of the complex.” Should be changed to “Epimeric compound pairs C15-C16C17-C18C19-C20, were explored to determine if the electron density of the proximal benzene ring has an effect on the efficiency of the complex.” 
  • Page 4 line 141: “To assess in the backbone chirality” should be changed to “To assess if the backbone chirality”
  • Page 7 Line 203/204: When you say, “The increase of the reaction temperature accelerated the transformation, but we also observed a small but perceptible erosion of the selectivity in both cases.” – what does this say about the kinetics of this transformation? 
  • Introduction could do with a figure, it seems as if there are no figures in this paper and just a bunch of charts and data. 
  • It is stated that neither the position, nor the electron withdrawing-donating nature of the system correlated systematically with the observed ee values. What might be causing this trend?
  • ESI: In some cases, there is experimental data missing – when discussing NMR (2.1 and 2.2)
  • What is a boom time? This sounds quite colloquial…

Author Response

We thank the thorough refereeing of our manuscript and send our answers below.

The authors should provide more reason as to why this is interesting and important… - the ability of this transformation to generate structural complexity was emphasized both in the introduction and discussion sections.

All specific proposals for the correction of the text were accepted and completed so they are not listed below.

  • Page 2 line 47: “ This generated a so far only partially met need for the stereo controlled generation of polysubstituted cyclopropanes and cyclobutanes” seems a bit wordy maybe change this up a bit. - changed to "In spite of this interest the stereo controlled generation of polysubstituted cyclopropanes and cyclobutanes remains challenging."
  • Page 2 line 73: “The progress of the reactions was monitored up to 14 days.” How were these reactions monitored? - we used chiral HPLC analysis of the reaction mixture, text was corrected to include this information
  • Page 7 Line 203/204: When you say, “The increase of the reaction temperature accelerated the transformation, but we also observed a small but perceptible erosion of the selectivity in both cases.” – what does this say about the kinetics of this transformation? - in the absence of detailed kinetic measurements we would not venture into further speculations, while the general acceleration of the transformation and the erosion of enantioselectivity at higher reaction temperatures seemed too trivial to explain.
  • Introduction could do with a figure, it seems as if there are no figures in this paper and just a bunch of charts and data. - There are several schemes in the manuscript. Figure 1 (Introduction) and Figure 2 (results and discussion), while the following 4 schemes are incorporated into the respective tables
  • It is stated that neither the position, nor the electron withdrawing-donating nature of the system correlated systematically with the observed ee values. What might be causing this trend? - we can't explain this finding and we though it more prudent to state these facts and not to present some weakly founded hypothesis. 
  • ESI: In some cases, there is experimental data missing – when discussing NMR (2.1 and 2.2) - corrected
  • What is a boom time? This sounds quite colloquial… - the introduction was corrected

Reviewer 3 Report

The paper titled ''Enantioselective cyclopropanation catalyzed by gold(I)-carbene complexes'' by Sazbo et. al. presents the investigation of chiral-NHC's complexed to Au(I) as catalysts for enantioselective cyclopropanation reactions. An extensive library of chiral NHC's has been tested with different conditions such as catalyst loading, solvents, additive, counter anions. The control experiments are done properly and adequtely. I am quite impressed by the honest and transparent conclusions drawn by authors. It is quite common now-a-days that many authors tend to oversell their results and highlight best ee obtained in catalysis. I am happy that authors of this paper are frank about the conclusion that they were not able to establish a strucutre-reactivity relationship in catalysis. I hope their future efforts in this direction may give more sucess in releasing such goals.

I recommend to accept the paper in Molecules after minor revisions. Here are my suggestions that need to be considered before publication:

1) There are few typos in the manuscript please revise carefully once more.

2) In the supporting information, please add the procedure for the synthesis and characterisation data (NMR) of all the new chiral NHC ligands. I would highly recommend to add a general procedure and 1H NMR (only data, spectra not needed) of already reported ligands as well and cite the relevent paper. This suggestions is to make it easy for readers to find a procedure easily, if they are interested in your complexes and ligands. Citing the ligand just in main manuscript is not enough.

3) Some of the silver compounds are not very pure and I can see some un-assigned impurities in the aromatic region in the provided figures. Please explain why they are not pure and mark the signals of impurity in the figures and try to assign the impurity if possible. Label all the signals corresponding to the residual proton signals in 1H and 13C{1H}. Also, for the whole paper 13C{1H} is mentioned just as 13C, there is a huge difference in both, please correct it.

4) All of the complexes are only characterised by NMR and HRMS. Why elemental analysis of any of the compound is not given? How did authors confirmed the absence of inorganic impruities like Au(I), Au(0) and Ag(I)? At least for one most successful catalyst they should check the elemental analysis.

5) If the samples are indeed contaminated, it could be that the Ag(I), Au(I) or Au(0) un-complexed to NHC also catalyze the cyclopropanation achirally and led to lower enantioselectivity. This could be checked by test reaction for cyclopropanation just with Au(I) salts. If the purity is proved by elemental analysis (comment 4) then comment 5 can be neglected.   

Author Response

We thank the thorough refereeing of our manuscript. Our answers and actions are listed below.

1) There are few typos in the manuscript please revise carefully once more. - completed

2) In the supporting information, please add the procedure for the synthesis and characterisation data (NMR) of all the new chiral NHC ligands. I would highly recommend to add a general procedure and 1H NMR (only data, spectra not needed) of already reported ligands as well and cite the relevent paper. This suggestions is to make it easy for readers to find a procedure easily, if they are interested in your complexes and ligands. Citing the ligand just in main manuscript is not enough. - we have amended the supporting information by including the general synthesis protocol for the already described ligand precursors A1-A40, and the synthesis and full characterization of the new NHC ligand precursors A41 and A42. 

3) Some of the silver compounds are not very pure and I can see some un-assigned impurities in the aromatic region in the provided figures. Please explain why they are not pure and mark the signals of impurity in the figures and try to assign the impurity if possible. Label all the signals corresponding to the residual proton signals in 1H and 13C{1H}. Also, for the whole paper 13C{1H} is mentioned just as 13C, there is a huge difference in both, please correct it. - the impurity of the Ag-salts comes from to the incomplete separation of the NHC-precursors, which are prepared as diastereoisomeric mixtures. Since their conversion to the Au-NHC-complexes and the subsequent purification resulted in homogeneous complexes we accepted the presence of the impurity in some of these silver intermediates. The label 13C{1H} was corrected throughout the manuscript. Where signals belonging to the minor diastereoisomer were clearly distinguishable they were denoted by an asterisk in the 1H spectra.

4) All of the complexes are only characterised by NMR and HRMS. Why elemental analysis of any of the compound is not given? How did authors confirmed the absence of inorganic impruities like Au(I), Au(0) and Ag(I)? At least for one most successful catalyst they should check the elemental analysis. - the elementary analysis of complexes C35 and C36 was measured and included in the Supporting Information. The purification and the EA results exclude the presence of residual Au-salts in a significant amount.

5) If the samples are indeed contaminated, it could be that the Ag(I), Au(I) or Au(0) un-complexed to NHC also catalyze the cyclopropanation achirally and led to lower enantioselectivity. This could be checked by test reaction for cyclopropanation just with Au(I) salts. If the purity is proved by elemental analysis (comment 4) then comment 5 can be neglected. - the elementary analysis results made these experiments unnecessary

Round 2

Reviewer 2 Report

Publication is recommended.